# Exploring the Structure and Substance Metabolism of a *Medicago sativa* L. Stem Base

**DOI:** 10.3390/ijms25116225

**Published:** 2024-06-05

**Authors:** Qian Gao, Kun Wang, Jing Huang, Pengpeng Dou, Zhengzhou Miao

**Affiliations:** College of Grassland Science and Technology, China Agricultural University, No. 2 Yuanmingyuan West Road, Haidian District, Beijing 100107, China; 18811456401@163.com (Q.G.); huangjing0412@cau.edu.cn (J.H.); pengpengdou@cau.edu.cn (P.D.); wwangwang885@gmail.com (Z.M.)

**Keywords:** *Medicago sativa* L., anatomy, transcriptome, RNA-seq, metabolites, metabolomics

## Abstract

The stem base of alfalfa is a critical part for its overwintering, regeneration, and yield. To better understand the specificity and importance of the stem base, we analyzed the structure, metabolic substances, and transcriptome of the stem base using anatomical techniques, ultra-high performance liquid chromatography tandem mass spectrometry (UPLC-MS/MS), and RNA sequencing (RNA-seq), and compared it with stems and roots. The anatomical structure shows that the ratio of xylem to phloem changes at the base of the stem. A total of 801 compounds involved in 91 metabolic pathways were identified from the broadly targeted metabolome. Transcriptome analysis revealed 4974 differentially expressed genes (DEGs) at the stem base compared to the stem, and 5503 DEGs compared to the root. Comprehensive analyses of differentially accumulated compounds (DACs) and DEGs, in the stem base vs. stem, identified 10 valuable pathways, including plant hormone signal transduction, zeatin biosynthesis, α-Linolenic acid metabolism, histidine metabolism, carbon metabolism, carbon fixation in photosynthetic organisms, pentose phosphate pathway, galactose metabolism, and fructose and mannose metabolism. The pathways of plant hormone signal transduction and carbon metabolism were also identified by comparing the stem base with the roots. Taken together, the stem base of alfalfa is the transition region between the stem and root in morphology; in terms of material metabolism, its growth, development, and function are regulated through hormones and sugars.

## 1. Introduction

Alfalfa (*Medicago sativa* L.), widely cultivated in agricultural systems around the world, is renowned for its notable nutritional value and various environmental advantages [1,2,3]. Its significance extends beyond mere fodder; alfalfa plays a critical role in enhancing soil fertility, promoting biodiversity, and contributing to the sustainability of farming practices [4,5,6]. A deeper understanding of alfalfa’s biological functions is critical, particularly the role of the stem base—a key region vital to the plant’s health and vigor [7,8].

The stem base, serving as the crucial junction between the root and stem, plays an essential role in the plant’s physiology by facilitating the transport of nutrients, photosynthates, water, minerals, and organic compounds throughout the plant. This key region supports the plant’s growth, enhances its resilience to environmental stressors, and contributes significantly to its productivity [9,10,11,12]. In alfalfa, the stem base is not merely a physical junction, but a vital hub of resilience, underpinning the plant’s remarkable ability to recover and regenerate [13]. This key area not only enables alfalfa to withstand environmental stressors and adverse conditions, but also supports its capacity to regrow vigorously after dormancy or harvesting [14]. These factors collectively influence the plant’s viability, productivity, ability to overwinter successfully, and the overall quality of the forage it produces [15,16,17].

The regenerative prowess of the stem base is critical for alfalfa’s endurance, enabling swift recovery and growth after harvesting, a vital factor for a consistent forage supply. This trait not only boosts the plant’s productivity, but also ensures the forage remains of superior quality, even in less-than-ideal growing conditions [18,19]. Furthermore, the stem base’s role is pivotal for successful overwintering, equipping alfalfa to navigate and flourish amid diverse climatic challenges [15,17]. These aspects highlight the stem base’s indispensable contribution to alfalfa’s survival, yield, and forage quality. Therefore, delving into the stem base’s functions emerges as a crucial area of study, promising to unlock strategies for enhancing alfalfa’s resilience and productivity.

Understanding the anatomical and metabolic intricacies of the stem base is vital for crafting strategies that improve alfalfa’s efficiency in resource use and its resilience to stress, insights that could also illuminate the stem bases of other plant species. Investigating this crucial region’s role within alfalfa’s lifecycle not only underscores its significance for the plant’s health and productivity, but also emphasizes its contribution to the sustainability of alfalfa as an essential agricultural commodity [19,20]. Such knowledge is instrumental in guiding breeding programs toward bolstering these key characteristics. Furthermore, the lessons learned from alfalfa stem base research promise wider implications, offering strategies to enhance the resilience and yield of various crops, thereby highlighting the necessity of continued research in this area.

Despite the acknowledged importance of the alfalfa stem base, comprehensive studies delving into its anatomical and metabolic complexities remain sparse. Existing research has primarily centered on yield optimization and genetic improvement, leading to an underexplored area concerning the role of the stem base in the plant’s lifecycle, a critical aspect for its growth and development. It becomes imperative to harness advanced scientific methodologies to unravel the complexities residing within this crucial segment. The integration of anatomical techniques with metabolomics and transcriptomics represents a cutting-edge approach, offering profound insights into the structural and functional dynamics of the alfalfa stem base [21,22,23]. Anatomical studies, utilizing both traditional and innovative imaging techniques, provide a detailed map of the stem base’s internal structure, enabling researchers to visualize and understand the spatial organization of different tissue types and their potential roles in plant physiology, critical for the transport and distribution of water, nutrients, and photosynthates [24,25,26]. Metabolomics complements anatomical studies by mapping the metabolites specific to cellular activities, revealing distinct metabolic pathways in the stem, stem base, and roots. This method identifies unique metabolic profiles across these regions, elucidating their specific roles in growth, stress response, and development. Similarly, transcriptomics provides a snapshot of gene expression within the stem base, revealing the genetic underpinnings of its anatomical and metabolic characteristics. High-throughput RNA sequencing allows for the identification of genes associated with critical functions, such as nutrient transport, signal transduction, and response to environmental stimuli [27,28,29]. By correlating gene expression patterns with anatomical and metabolic traits, researchers can decipher the complex molecular mechanisms governing the stem base’s contribution to alfalfa’s growth and productivity.

In this study, our goal is to seamlessly blend traditional anatomical methods with advanced genomic, transcriptomic, and metabolomic analyses to explore the intricate structural and metabolic interplay within the alfalfa stem base, which is vital for the plant’s growth and resilience. By elucidating the complex structural and functional dynamics of this essential region, we aim to identify key metabolic pathways and uncover the genetic underpinnings that support its critical functions. This comprehensive understanding is expected to bridge a significant research gap in alfalfa studies, paving the way for the development of enhanced cultivation practices and breeding strategies that could have wide-reaching implications for plant science and agricultural innovation.

## 2. Results

### 2.1. Identification of Metabolites in Three Plant Tissues

The regenerative capacity of alfalfa’s stem base is an important characteristic of alfalfa, and the stem base is also the transition zone between root and stem, so in our study, we divided alfalfa plants into three parts—stems, stem base, and roots—to explain the special position of the stem base in alfalfa growth by analyzing the differences between anatomy, metabolomes, and transcriptomes among different alfalfa tissues (Figure 1a). To explore the differential metabolites in the stem base, the broadly targeted metabolome was performed on plant tissues from three parts. A total of 801 compounds were identified, and these metabolites can be broadly classified into 11 types, namely flavonoids (21.35%), lipids (12.98%), phenolic acids (11.99%), terpenoids (10.49%), amino acids and derivatives (9.49%), alkaloids (6.99%), organic acids (5.87%), nucleotides and derivatives (5.12%), lignans and coumarins (3.12%), quinones (1.12%), and others (11.49%) (Appendix A). These compounds were involved into 91 pathways, mainly metabolic pathways, biosynthesis of secondary metabolites, ABC transporters, biosynthesis of cofactors, biosynthesis of amino acids, and linoleic acid metabolism, among others (Figure 1b).

### 2.2. Structure Analysis

The anatomical structure provides a visual indication of the differences that exist between the stem, stem base, and roots (Figure 2). The distribution of xylem (amaranthine red parts) varies very markedly in these three parts, from rays in the roots to a ring-like shape in the stems, with a transition occurring at the stem base—the staining showed that the outer xylem cells were rayed and the inner ones showed an annular morphology. The same phloem is also altered, with the phloem in the roots arranged interspersed with the xylem, and the phloem in the stems ringed like the xylem, with the stem base likewise exhibiting a transitional form. In addition, the stem has a hollow medullary cavity, and the stem base with the root is a complete solid structure.

The software CaseViewer 2.0 was utilized to extract xylem and phloem areas and calculate the ratio of xylem to phloem. As depicted in Figure 3, the distribution of xylem to phloem in the stem is approximately 1:1. However, toward the stem base, the xylem is approximately 2/3 the size of the phloem. In the root, the xylem area is less than half of the phloem area.

### 2.3. Differential Metabolites in the Stem Base

Principal component analysis of the relative content of compounds showed that three replicates of each of the sample groups clustered together, indicating the similarity between biological replicates within each group. The first and second principal components (PC1 and PC2, respectively) of the sample set data accounted for 37.46% and 30.01% of the total variability, respectively (Figure 4a). The PCA results indicate that the stem, stem base, and root have relatively large overall metabolic differences. Among them, the samples of stems and roots were very well reproducible, while the stem base showed separation in the second principal component. The PCA results indicate that the stem base differs from the stem and root in terms of metabolites (Appendix A).

Metabolites were screened for up-regulation at the stem base compared to stems and roots, then classed according to their origins and chemical characters (Figure 4b,c). The metabolites that were up-regulated at the stem base compared to the stem were classified into nine superclass, and the three categories containing the most metabolites were flavonoids, amino acids and derivatives, and alkaloids. Subdividing these metabolites into 22 subclasses, amino acids and derivatives, flavones, and triterpene saponin were the categories that contained the largest variety of metabolites.

The metabolites up-regulated at the stem base compared to the roots were classified into 11 superclass, and the three categories containing the most metabolites were flavonoids, phenolic acids, and other (containing 20 sugars). Subdividing these metabolites yielded 22 subclasses, with phenolic acids, flavones, and saccharides being the classes containing the largest variety of metabolites.

### 2.4. Identification of Differentially Expressed Genes

The differentially expressed genes (DEGs) in the three parts of the tissue were investigated by transcriptome sequencing (Table 1). A total of 15,763 genes were detected from plant samples. Among these genes, 1814 genes were up-regulated and 3160 genes were down-regulated at the stem base compared to the stem, and at the stem base compared to the root, 3243 genes were up-regulated and 2260 genes were down-regulated (Appendix A).

The classification of GO enrichment suggests that the DEGs were classified into molecular function, biological process, and cellular component, and the top 20 with the smallest *p*-value as shown in the figure (Figure 5a). The differences between stem base and stems are mainly related to photosynthesis, including chlorophyll synthesis and metabolism-related processes, as well as some cell wall development pathways, whereas DEGs of the stem base and roots were enriched in the regulation of hormone levels and glucose metabolic activity.

The KEGG enrichment pathway of DEGs from the stem base and stem is different from that of the stem base and root (Figure 5b). In the comparisons of the stem base and stem, the plant hormone signal transduction pathway, zeatin biosynthesis pathway, and glycolysis/gluconeogenesis pathway were enriched. Meanwhile, the photosynthesis pathway was also detected, together with several other pathways related to photosynthesis. In the comparisons of the stem base and root, the starch and sucrose metabolism pathway, carotenoid biosynthesis pathway and alpha-linolenic acid metabolism, and plant hormone signal transduction pathway were enriched.

The analysis revealed significant differences in metabolic pathways between the stem base and the stem, with notable emphases on photosynthesis, sugar metabolism, cell wall development, and hormone signaling pathways. Similarly, compared to the roots, significant differences were observed in hormone metabolism, carotenoid synthesis, and starch and sucrose metabolism.

### 2.5. Association Analysis between DEGs and DACs

Combining the transcriptome and metabolome results, the 25 pathway entries with the most significant enrichment were screened (Figure 6). For technical reasons, certain metabolites could not be detected by widely targeted metabolomics techniques, so we focused primarily on transcriptomics results when performing our co-analysis.

The results of the conjoint analysis of DACs and DEGs in the stem base and stem are shown in Figure 6a. There are four pathways associated with hormones, namely plant hormone signal transduction, zeatin biosynthesis, α-Linolenic acid metabolism, and histidine metabolism. The zeatin biosynthesis pathway precedes cytokinin synthesis, histidine is the catalyzing enzyme for cytokinin synthesis, and jasmonic acid is derived from linoleic acid (Figure 6c). Compared to the stem, in plant hormone signal transduction, almost all the genes of the auxin pathway, cytokinin pathway, gibberellin pathway, brassinosteroid pathway, jasmonic acid pathway, and salicylic acid pathway were down-regulated at the stem base, and some genes for zeatin biosynthesis in the upstream of cytokinin pathways are also down-regulated, whereas half of the genes of the abscisic acid pathway and ethylene pathway were up-regulated (Appendix A). There are five pathways associated with energy substances, which are carbon metabolism, carbon fixation in photosynthetic organisms, the pentose phosphate pathway, galactose metabolism, and fructose and mannose metabolism. These 5 pathways were enriched for a total of 12 substances and 197 genes, and 11 substances were at lower levels in the stem base than the stem, and most genes were down-regulated for expression (Appendix A).

The results of the conjoint analysis of DACs and DEGs at the stem base and root are shown in Figure 6b. Several key saccharides, glucose-1-phosphate, D-glucose, and D-sucros, link six energy–substance metabolic pathways (Figure 6d). Glucose is higher at the base of the stem compared to the root, while sucrose and glucose-1-phosphate are lower. Regarding the transcriptome results, some genes involved in carbon fixation and carbon metabolism pathways are up-regulated at the stem base. The plant hormone signaling pathway is also enriched in the stem base, but the regulatory trends of the genes do not converge (Appendix A).

### 2.6. Validation of Differential Gene Expression

To validate the accuracy and reproducibility of the RNA-seq expression results, we selected 7 DEGs and designed primers for real-time PCR detection. We observed that the FPKM value had a similar tendency to the relative expression levels (Appendix A), which further confirmed the reliability of our transcriptome data.

## 3. Discussion

In agricultural sciences, understanding the anatomical and physiological features of crops like alfalfa is essential for improving productivity and quality. The stem base, where the root system transitions into the stem, is vital for nutrient and water transport and plays a key role in plant stability and disease resistance. The robustness of the stem base enhances alfalfa’s resilience to environmental stresses, which impacts crop health and yield. Therefore, strengthening the structural integrity and functional capacity of the stem base could be a novel approach to boosting alfalfa’s productivity and quality.

### 3.1. Transport Channels—Xylem and Phloem

The xylem and phloem, essential for nutrient and water transport in plants, closely interact and coordinate [30,31]. Roots absorb water and nutrients, transported through the xylem to leaves for photosynthesis, while the phloem uses transpiration-generated forces to transport sugars to sink organs. Measuring phloem transport directly is challenging due to phloem fragility [32]. In this study, we estimate substance transduction roughly by measuring xylem and phloem areas.

Figure 2 illustrates the structure of alfalfa, specifically the rayed structure of the root xylem, which shortens the distance for water and nutrient entry into the plant transport system (xylem). The stem base exhibits a transitional form between the root and stem, with xylem rayed like the root but tending to develop into an annulus. Branching is a fundamental function of the alfalfa stem base. The anatomical structure reveals that branches appearing at different times exhibit distinct staining patterns in the cross-section, with early appearing branches aligning closer to the root pattern and late-appearing branches situated closer to the stem pattern. Progressing toward the stem, the phloem is organized in a ring adjacent to the cortex, effectively accepting the sugars produced by the leaves and functioning as an efficient transit channel.

In addition to visual morphological observations, we calculated the xylem-to-phloem ratio (x/p), revealing that the stem has the largest x/p (9/10), the base stem has an intermediate x/p (3/5), and the root has the smallest x/p (1/3). This outcome might be roughly interpreted as a gradual enlargement of the xylem proportion from the root to the stem and a corresponding increase in the phloem proportion from the stem to the root. The distribution of xylem and phloem is designed for efficient material transport. The stem, with its substantial xylem, continuously draws water and nutrients from the roots for photosynthesis. The larger phloem, located in the lower part of the plant, efficiently transports the sugars synthesized in the upper part to sustain metabolic activity across various tissues in the body.

It is fascinating to observe that the arrangement of a plant’s xylem and phloem is consistently structured to facilitate the rapid transport of nutrients, adhering to the principle of maximizing transport efficiency. Plants inherently exhibit efficiency in their forms, organs, and functions. Considering their lack of locomotion compared to animals, every developmental aspect must be optimized to maximize benefits. The arrangement and size of the xylem and phloem alone signify that plants utilize the space within their bodies in a skillful manner.

### 3.2. The Regenerative Capacity of Alfalfa Stem Bases

Our experiment aimed to dissect the transcriptomic divergence among the stem, stem base, and root of *Medicago sativa* L., focusing on hormonal pathways and carbohydrate metabolism to understand tissue-specific regulatory mechanisms and elucidate the stem base’s unique functional roles in the plant. Subsequently, we identified numerous candidate pathways and genes implicated in the development of the alfalfa stem base.

Research has elucidated the critical role of phytohormones in plant growth, development, and adaptation to environmental challenges. These hormones, even in trace amounts, exert profound regulatory effects, working synergistically to modulate a diverse range of developmental processes and adaptive responses to environmental stimuli [24,25,26]. Cell division, differentiation, and elongation are fundamental to the regenerative capacity of the alfalfa stem base, driven by a complex interplay of phytohormones, such as CK, IAA, brassinosteroids (BRs), and gibberellins (GAs) [33,34,35]. This intricate regulation involves a network of genes, including the A-type ARABIDOPSIS RESPONSE REGULATORS (ARRs) for cell division [36], auxin response factors (ARFs) for IAA response [37], DELLA proteins for growth restraint [38], and the TCP family, which comprises key transcription factors for bud differentiation and sprouting [39].

In our gene expression analysis, we selected genes with a fold change (FC) threshold greater than 2 and adjusted the false discovery rate (FDR) to identify significantly differentially expressed genes (Appendix A and Figure 7). Our findings reveal that all DEGs associated with CK signaling are down-regulated at the stem base. Furthermore, within the BR signaling pathway, certain hormone receptor kinases, including the brassinosteroid insensitive 1 (BRI1) protein, and the cell cycle regulator, cyclin D3, were found to be down-regulated in comparison to both stem and root tissues. This suggests that cell division and elongation at the stem base may be slower, possibly indicating a dormant state. IAA does not directly influence regeneration, but exerts its effects through two secondary messengers: CK and SL [40]. As a result, in the stem base, the DEGs associated with the IAA signaling pathway are predominantly down-regulated, although some genes involved in its biosynthesis exhibit up-regulation. In the GA signaling pathway, DELLA proteins act as inhibitors of cell elongation and division, thus limiting plant growth. Even though the expression levels of DELLA proteins were reduced at the stem base, the presence of these proteins in any amount continued to inhibit plant growth.

The TCP family of transcription factors plays pivotal roles in the regulation of plant growth and development, exerting control over key processes such as cell proliferation, differentiation, and hormone signaling pathways [41,42]. In this study, we detected the differential expression of seven members of the TCP transcription factor family at the stem base, as shown in Figure 8. These include the candidate homolog of BRC1 in alfalfa (MsG0880046617.01), with an 82% sequence similarity to its counterpart in *Pisum sativum*.

Phylogenetic scrutiny of the TCP family across a swath of plant taxa—encompassing dicots and monocots—underscored a remarkable evolutionary constancy within legumes, buttressed by strong bootstrap support. Notable divergence among more evolutionarily removed species suggests an evolutionary adaptation and functional divergence. The consistent clustering of *Arabidopsis thaliana*’s BRC1 with legume TCPs underscores their preserved regulatory role in plant structural development across diverse plant families. This phylogenetic insight furnishes a foundation for future research into the developmental significance of TCP transcription factors in plants.

Recent studies have identified changes in plant carbohydrate content, particularly soluble sugars, during plant lateral regeneration [43]. Barbier et al. found that HXK1 in rose, the first enzyme of glycolysis, is a key signaling component in shoot branching, in the sugar signaling network [44]. We found several pathways in comparative analyses of transcriptome data that suggest a linkage between carbohydrate metabolism and stem base. For example, the enrichment of two KEGG pathways related to ‘Fructose and Mannose metabolism’ and ‘Glycolysis/Gluconeogenesis’ further underscores the significant role of sugar metabolism at the stem base. Building on prior research [45], we hypothesize that sugars located at the stem base serve not only as sources of energy and nutrients, but also as signaling molecules that influence hormonal balances and gene expression patterns. Studies targeting HXK1 have also found that it interacts with the signaling pathways of CK, IAA, and SL to regulate shoot branching [44]. Hence, our data lead us to hypothesize that sugars and hormones are integral to the regenerative capacity of alfalfa stem bases, a hypothesis that will be rigorously tested in our subsequent investigations.

## 4. Materials and Methods

### 4.1. Plant Materials

The alfalfa variety used in this study was Zhongmu No. 1, grown at the Shang Zhuang Experimental Station (116°10′44.83″ E, 40°08′12.15″ N) at China Agricultural University for 3 years. The test station experiences an average annual precipitation of about 595 mm and an average annual temperature of approximately 12 °C. We sampled during the vigorous growth stage of alfalfa, when the plants were approximately 1.5 m tall and in the early flowering period. Five alfalfa plants with consistent growth were selected from the experimental field, uprooted together with their roots. After comparing the integrity and growth status of the plants, the most intact and least damaged plant, with minimal pest infestation, was chosen for subsequent experiments. The stems (s), stem base (c), and root (r) of the plants were collected for experiments. The samples were cleaned, quick-frozen with liquid nitrogen, and then stored in a −80 °C refrigerator for subsequent experiments.

### 4.2. Histological Analysis

Each of the three parts of the sample was fixed in formaldehyde-acetic acid-ethanol (FAA) fixative (70%); paraffin sections (10 µm) were created after 1 month of storage. We used xylene and anhydrous ethanol to deparaffinize paraffin sections. The sections after rinsing were immersed in saffron dye for 1–2 h. The excess dye was washed away with water and the sections were then decolored by immersing them in gradients of 50%, 70%, and 80% alcohol. Then the sections were immersed in fast green solution for 30–60 s, then washed with anhydrous ethanol and dehydrated. The stained sections were put into n-butanol and xylene sequentially for 5 min each, taken out to dry, and sealed with neutral resin. Sections were observed under a microscope for image acquisition and analysis (scanner model: Pannoramic 250/MIDI, 3DHISTECH, Budapest, Hungary).

### 4.3. Extraction and Measurement of Metabolites

The samples were vacuum freeze-dried in a lyophilizer (Scientz-100F, Scientz, Ningbo, China) and then ground (30 Hz, 1.5 min) to powder form using a grinder (MM 400, Retsch). Samples (50 mg) were extracted using 1 mL of 70% methanol. The extracts were vortexed six times to accelerate the extraction rate (every 30 min for 30 s). The samples were centrifuged for 3 min (at 12,000 rpm), and the supernatant was aspirated. The samples were filtered through a microporous membrane (0.22 μm pore size) and stored in an injection vial for subsequent UPLC-MS/MS analysis. The data acquisition instrument system mainly included ultra-performance liquid chromatography (UPLC) (ExionLC™ AD, https://sciex.com.cn/, (accessed on 2 October 2023)) and tandem mass spectrometry (MS/MS) (Applied Biosystems 4500 QTRAP, https://sciex.com.cn/ (accessed on 23 October 2023)). Three independent biological replicates were conducted for each sample.

### 4.4. Transcriptome Sequencing

First, total RNA was extracted from three parts of the plant samples, and the RNA quality was tested. Then, library preparation for transcriptome sequencing was performed, after the library was constructed, the quality of the library was tested, and the results met the requirements before sequencing was performed on the machine. Transcriptomic sequencing was performed by Wuhan Metware Biotechnology Co., Ltd., Wuhan, China; the specific operation can be found on the online platform (https://cloud.metware.cn (accessed on 26 May 2024)). Use fastp v 0.19.3 to filter the original data, mainly to remove reads with adapters; when the N content in any sequencing read exceeds 10% of the base number of the reads, remove the paired reads; when any sequencing read when the number of low-quality (Q ≤ 20) bases contained in the reads exceeds 50% of the bases of the reads, this paired read will be removed. All subsequent analyses are based on clean reads. The reference genome and its annotation files are provided in the Appendix A (Muxu_Genome_Final.fasta); we used HISAT to construct the index and compare clean reads to the reference genome [46]. Use StringTie for new gene prediction. StringTie applies network streaming algorithms and optional de novo to splice transcripts. Compared with Cufflinks and other software, StringTie [47] can splice a completer and more accurate transcript, and the splicing speed is faster. Use featureCounts [48] to calculate the gene alignment, and then calculate the FPKM of each gene based on the gene length. FPKM is currently the most used method to estimate gene expression levels. DESeq2 [49,50] was used to analyze the differential expression between the two groups, and the *p*-value was corrected using the Benjamini and Hochberg method. The corrected *p*-value and |log2foldchange| were used as the threshold for significant difference expression. Enrichment analysis was performed based on the hypergeometric test. For KEGG, the hypergeometric distribution test was performed with the unit of pathway; for GO, it was performed based on the GO term.

All the clean reads have been deposited in the National Center for Biotechnology Information (NCBI) Short Read Archive (SRA) Sequence Database under accession number PRJNA1045835.

### 4.5. Real-Time RT-PCR

The primer sequences used in this study were designed using Primer Premier 5.0 software, as listed in Appendix A. MsActin was selected as the reference gene for data normalization. Real-time PCR reactions were performed using the same RNA samples from the RNA-seq analysis. Relative gene expression levels were estimated using the threshold cycle method [51].

### 4.6. Data Analysis

Principal component analysis was performed using two-dimensional principal component analysis (PCA). Student’s *t*-test was served for analysis of significant difference. Analyses and mapping were performed using the Metware Cloud, a free online platform for data analysis (https://cloud.metware.cn (accessed on 2 November 2023)).

The software for the histological analysis was caseviewer2.0.

## 5. Conclusions

In this study, we employed a comprehensive approach, integrating photomicrographic techniques, transcriptomics, and metabolomics, to scrutinize the pivotal functions of the stem base in Zhongmu No. 1. The anatomical examination revealed a noteworthy transformation in the primary transport channels—xylem and phloem—occurring at the base stem. This morphological shift, from a ray-like to a ring-like structure, was accompanied by changes in area. Through the amalgamation of metabolomics and transcriptomics analyses, we identified differential enrichment in metabolic pathways, including those related to plant hormone signal transduction, zeatin biosynthesis, carotenoid biosynthesis, as well as starch and sucrose metabolism pathways. Consequently, we posit that the base stem of alfalfa represents a critical zone for transport channel transformation and branching genesis. Notably, energy substance metabolism and hormone signaling emerged as predominant factors influencing branching, as illustrated in Figure 9.

While our study has unveiled a series of crucial findings in this domain, it merely marks the initiation of a comprehensive exploration into the myriad mysteries surrounding plant growth and development. These discoveries not only offer novel perspectives for our understanding of plant physiology, but also furnish valuable cues for future investigations. We anticipate that these findings will form a robust foundation for practical applications in agriculture, plant breeding, and allied fields.

Nevertheless, we acknowledge that our study comes with certain limitations and unresolved questions. Future research can delve deeper, exploring more intricate molecular mechanisms to unravel further details of how the stem base regulates branching.

## Figures and Tables

**Figure 1 ijms-25-06225-f001:**
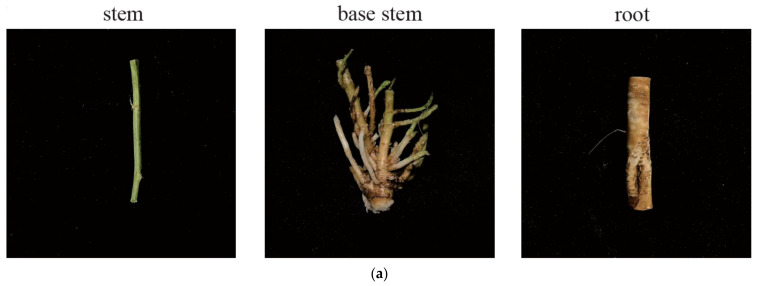
Metabolite class composition. (**a**) The plant material of *Medicago sativa* L. (**b**) Metabolite analysis of plant tissue.

**Figure 2 ijms-25-06225-f002:**
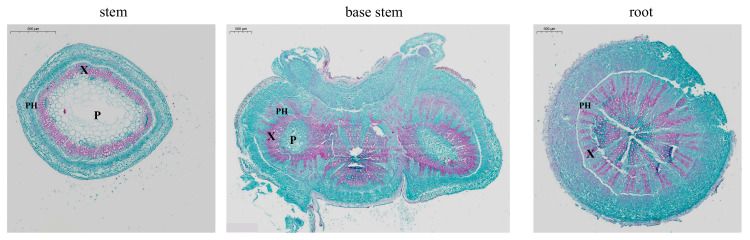
Transverse sections of stem, stem base, root. Phloem (PH) in xylem (X) and large pith (P), amaranthine red (lignification).

**Figure 3 ijms-25-06225-f003:**
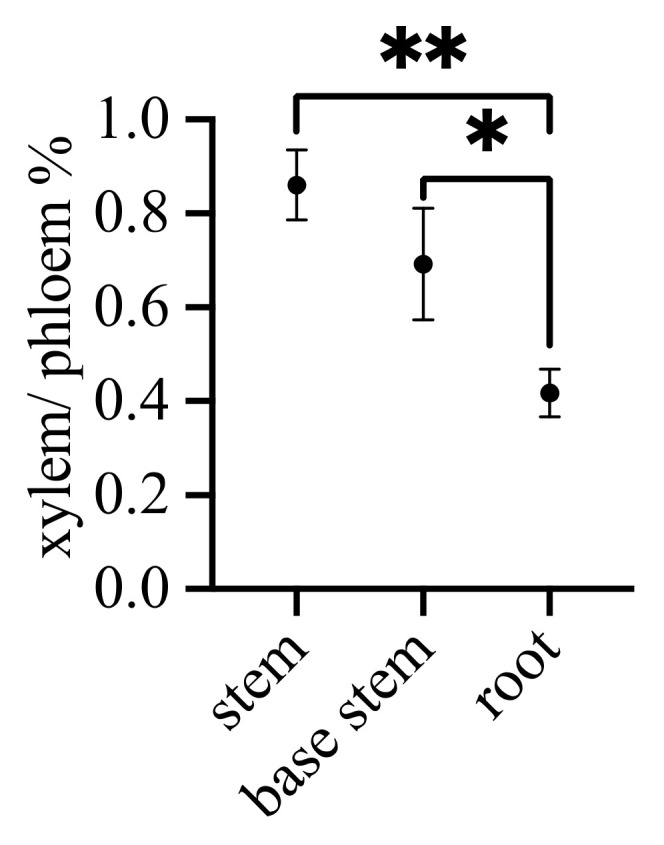
The ratio of xylem to phloem area. Note: * *p* < 0.05, ** *p* < 0.01.

**Figure 4 ijms-25-06225-f004:**
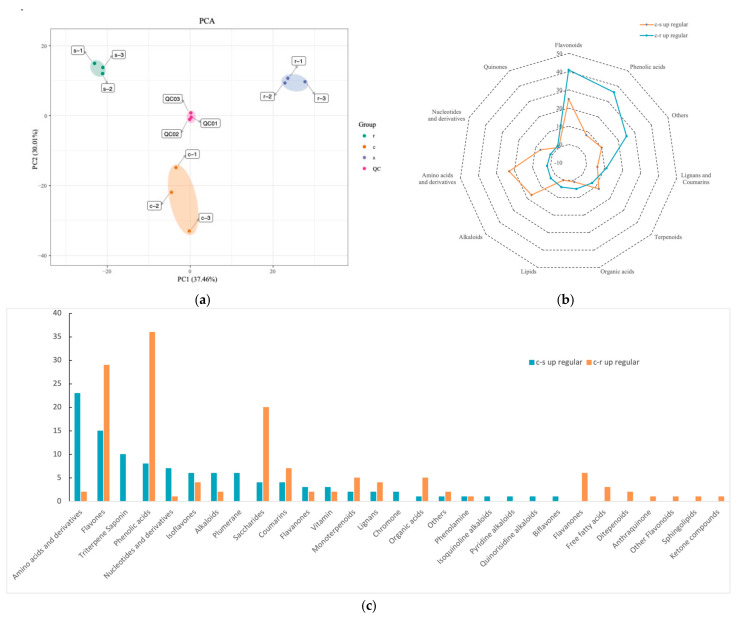
Differentially accumulated analysis of the isolated compounds between three tissues. (**a**) PCA suggested the accumulation of compounds in different tissues. (**b**) Enrichment of differentially accumulated compounds into large classes of metabolic pathways. (**c**) Enrichment of differentially accumulated compounds into subclass metabolic pathways.

**Figure 5 ijms-25-06225-f005:**
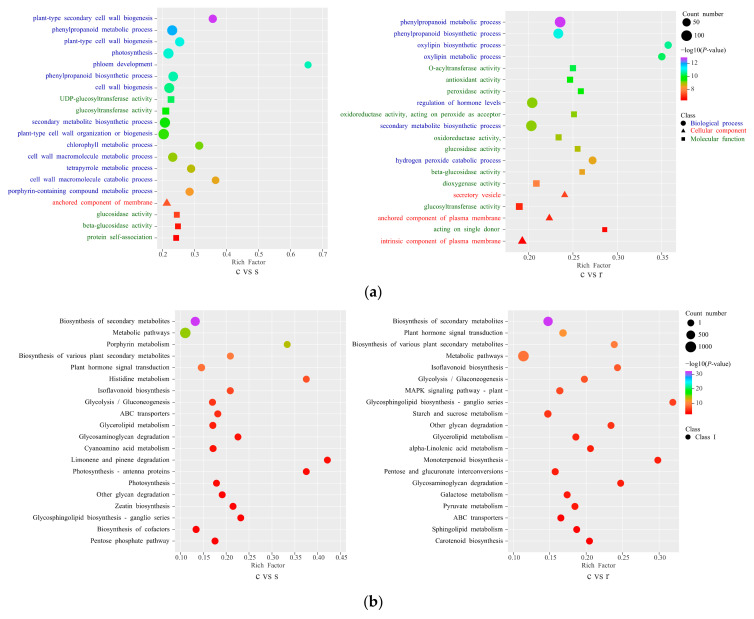
GO and KEGG enrichments of the differentially expressed genes (DEGs) between three tissues. The union set of 20 pathways with the smallest q value for each comparison combination is taken for presentation. (**a**) Scatterplot of multi-combination GO enrichment. (**b**) Scatterplot of multiple combinations of KEGG enrichment.

**Figure 6 ijms-25-06225-f006:**
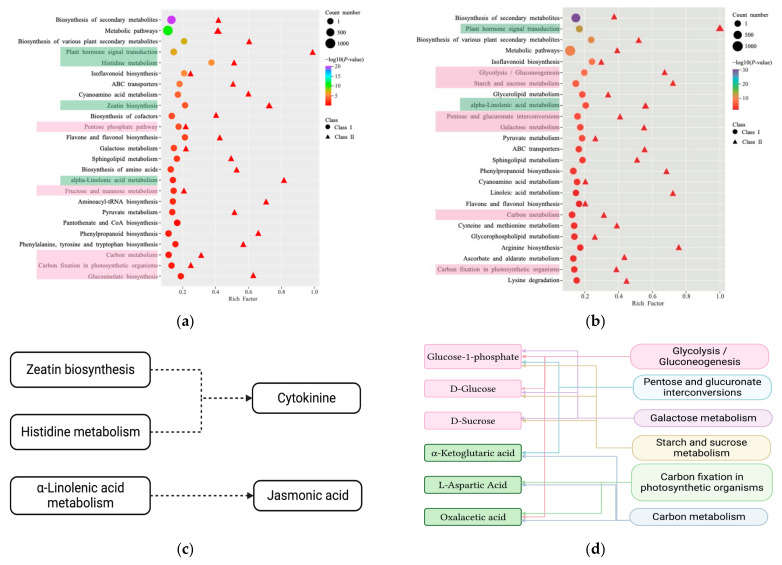
Combined transcriptome and metabolome analysis. (**a**) KEGG enriched bubble plots of stem base vs. stem. (**b**) KEGG enriched bubble plots of stem base vs. root. (**c**) Schematic representation of the association of hormone-related pathways. (**d**) Schematic representation of the association of saccharides with the pathway. Pink is energy metabolism-related pathway; Green is the hormone-related pathway. The arrow direction is downstream of the synthesis.

**Figure 7 ijms-25-06225-f007:**
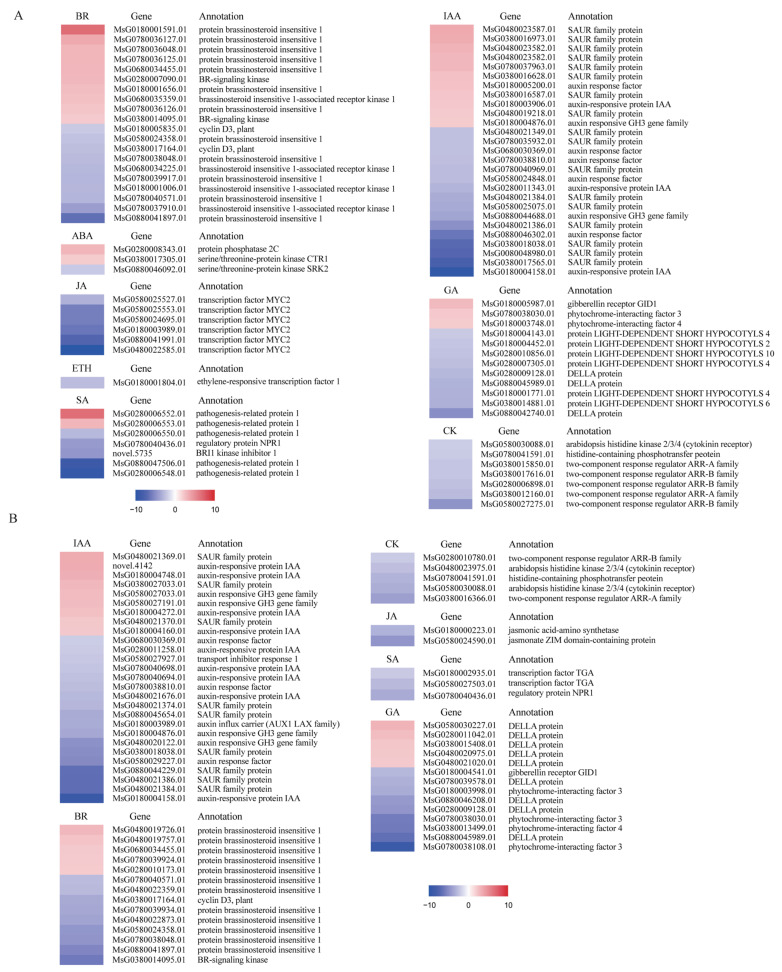
Heatmap diagram of relative gene expression levels of DEGs involved in phytohormone signal transduction. (**A**) Stem base vs. stem; (**B**) stem base vs. root. Blue indicates down-regulation, red indicates up-regulation, with darker shades representing higher significance. Gene annotations were performed using the NR (non-redundant) database. Novel gene annotations: 5735 in Chr5, (RefSeq) BRI1 kinase inhibitor 1 (A); 4142 in Chr3, (GenBank) auxin-responsive AUX/IAA family protein (A).

**Figure 8 ijms-25-06225-f008:**
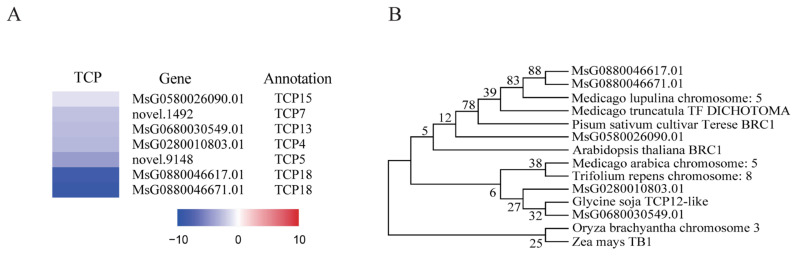
The differential expression of TCP members in stem base and the phylogenetic analysis. (**A**) Seven TCP members. (**B**) Phylogenetic tree of the homologous genes of MsG0880046617.01. All TCP members were down-regulated; the darker the color, the more significant the down-regulation. Gene annotations were performed using the NR (non-redundant) database. In the phylogenetic tree, each node on the tree represents a specific gene, with branch lengths proportional to genetic distances. Species include *Medicago lupulina*, *Pisum sativum*, *Arabidopsis thaliana*, *Medicago truncatula*, *Trifolium repens*, *Glycine soja*, *Oryza brachyantha*, and *Zea mays*. Novel gene annotations: 1492 in Chr1, (GenBank) TCP family transcription factor (A); 9148 in Chr7, (Swissprot) TCP5_ARATH RecName: Full = Transcription factor TCP5.

**Figure 9 ijms-25-06225-f009:**
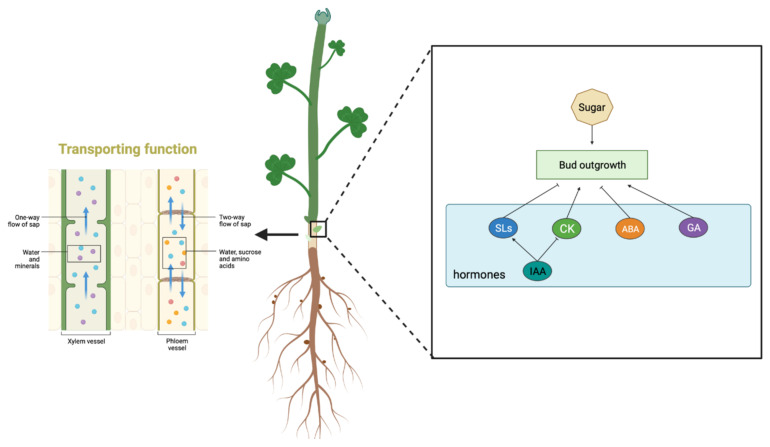
A schematic diagram of the function of the *Medicago sativa* L. stem base.

**Table 1 ijms-25-06225-t001:** The sequence reads of transcriptome and their mapping results to a reference genome.

Sample	Replicate	Raw Reads	Clean Reads	Reads Mapped	Clean Base (G)	Error Rate (%)	Q20 (%)	Q30 (%)	GC Content (%)
stem	s-1	46,525,180	44,653,312	35,696,908 (79.94%)	6.7	0.03	97.77	93.69	41.49
	s-2	49,274,168	47,298,816	36,331,319 (76.81%)	7.09	0.03	96.84	91.34	41.05
	s-3	44,999,472	43,015,244	33,554,924 (78.01%)	6.45	0.03	97.01	91.68	41.33
stem base	c-1	47,309,760	45,108,452	34,803,807 (77.16%)	6.77	0.03	97.02	91.88	41.43
	c-2	43,655,292	42,248,676	32,729,350 (77.47%)	6.34	0.03	96.71	90.99	41.28
	c-3	48,859,978	47,321,596	36,753,294 (77.67%)	7.1	0.03	96.84	91.29	41.28
root	r-1	47,351,972	45,320,980	34,880,708 (76.96%)	6.8	0.03	96.91	91.55	40.81
	r-2	44,140,624	42,271,582	32,883,539 (77.79%)	6.34	0.03	97.6	93.33	40.82
	r-3	52,453,492	50,657,998	39,267,720 (77.52%)	7.6	0.03	96.88	91.5	41.08

## Data Availability

Data are available from a publicly accessible repository.

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
