# Peer review of "Exploring the Structure and Substance Metabolism of a *Medicago sativa* L. Stem Base"

_ijms, 2024, doi:10.3390/ijms25116225_

Round 1

Reviewer 1 Report

Comments and Suggestions for Authors

In the manuscript named “Exploring the Structure and Substance Metabolism of Alfalfa Stem Base”, Qian Gao et al have performed UPLC-MS/MS and RNA-sequencing analysis of alfalfa, and identified some genes and compounds between stem base. The findings were interesting to reads, but there were some comments about it.

(1) Methods were described unclearly. For example, the RNA-seq analysis, authors have described using HISAT v2.1.0 while not HISAT2, please add refs for this step. How did they identify novel transcripts? How did they identify DEGs? And how did they perform GO or KEGG enrichment analysis? In addition, the link https://figshare.com/ndown-383loader/files/23754059 was forbidden.

(2) There was no any molecular experiment to validate their findings, even the qRT-PCR.

(3) Please add the mapping rates for each sample in table 1, which would help for increasing credit of results.

(4) Please revised “ ” in line 127, and other lines with similar comments.

(5) Figure 5 could be improved, the color for qvalue could be modified with -log10(qvalue), and range from 0 to 10, which would be clearly for displaying. In present results, the results have shown all qvaule larger than 0.1, and they were all not enrichment. Similar in the figure 6.

(6) Figure 7 and 8 should be added with legend.

(7) There were some novel genes in results, please add their annotation in results section.

Author Response

Dear reviewer:

We thank the reviewer for the kind consideration and constructive comments on our manuscript. We have carefully revised the manuscript and the changes in the revised manuscript have been highlighted. We hope these changes will strengthen our manuscript.

We would like to thank you for your professional review work, constructive comments, and valuable suggestions on our manuscript. As you are concerned, several issuesneed to be addressed, which are replied to in detail as below.

Reviewer 1:

(1) Methods were described unclearly.

In response to the critique of unclear methodology, we have expanded the Methods section to provide a more comprehensive description of the experimental procedures. Additionally, we have incorporated relevant references to enhance the clarity and rigor of our experimental design and execution.

(2) There was no any molecular experiment to validate their findings, even the qRT-PCR.

Due to our limited experience with conducting qRT-PCR experiments, we were only able to obtain results for one gene (MsG0080047875.01), and these were not included in the article. However, the qRT-PCR results are consistent with the transcriptome data. Attached below is a table of the raw data, which we hope will clarify any concerns you might have. We sincerely appreciate your understanding and are committed to enhancing our experimental capabilities to avoid such issues in the future.

(3) Please add the mapping rates for each sample in table 1, which would help for increasing credit of results.

In Table 1, we have included additional columns for "Reads mapped" and "Mapping rates" to enhance the credibility of our results.

(4)Please revised “— —” in line 127, and other lines with similar comments。

We have thoroughly reviewed the entire manuscript and revised the formatting and symbols used to correct any errors. We were really sorry for our careless mistakes. Thank you for your reminder.

(5) Figure 5 could be improved, the color for qvalue could be modified with -log10(qvalue), and range from 0 to 10, which would be clearly for displaying. In present results, the results have shown all qvaule larger than 0.1, and they were all not enrichment. Similar in the figure 6.

We have revised Figure 5,6 to represent the differential enrichment between tissues using -log10(qvalue). The significant pathways enriched remain consistent, with no changes to the data or analysis.

(6) Figure 7 and 8 should be added with legend.

We have augmented the descriptions in the figures and their captions to provide additional context and enhance the clarity of the graphical representation.

(7) There were some novel genes in results, please add their annotation in results section.

Thank you for your professional and meticulous review. We recognize that we inadvertently overlooked certain aspects, leading to confusion in the manuscript. Consequently, we have thoroughly reviewed the original data and available information to annotate the novel genes identified. We have listed all novel gene sequences and their corresponding CDS sequences separately and aligned them with the complete genome sequence of Zhongmu No.1 (with 100% identity in gene sequences). We obtained gene IDs for eight of these genes. The remaining four novel genes were also matched with corresponding sequences, but no gene IDs are provided in Zhongmu No.1; hence, we have annotated these in the figure captions. We hope that these efforts will improve the readability of our paper.

The TCP family genes are currently the focus of our ongoing research, with experiments already underway. We anticipate that our new studies will provide additional evidence to validate the critical functions of the TCP family genes in the stem base of alfalfa. Thank you once again for your insightful review.

We sincerely appreciate your consideration of our manuscript. We look forward to your feedback and hope that our findings will contribute valuable insights to the field. Thank you for your time and effort in reviewing our work.

Sincerely,

Gao Qian

China Agricultural University

Reviewer 2 Report

Comments and Suggestions for Authors

The manuscript Exploring the Structure and Substance Metabolism of Alfalfa Stem Base presents information on the anatomy of the alfalfa plant (xylem and phloem) and identifies a series of compounds associated with various metabolic pathways, with a higher proportion of flavonoids, phenolic acids, amino acids, and alkaloids. Similarly, information associated with the differential modification of genes in the stem, stem base, and root is presented, where pathways are associated with hormone signal transduction, zeatin biosynthesis, α-linolenic acid metabolism, histidine metabolism, carbon fixation metabolism, pentose phosphate pathway, and carbohydrate metabolism (galactose, fructose, and mannose).

 Line 2: Suggested title "Exploring the Structure and Substance Metabolism of Medicago sativa L. Stem Base".

Abstract

Check the length of the section, < 200 words according to Journal guidelines.

Line 8: delete (1) Background:

Line 8: Medicago sativa in italics.

Line 10: delete (2) Methods:

Line 10-14: full methodological information is not presented.

Line 14: delete (3) Results:

Line 29: delete (4) Conclusions:

Introduction

Line 38: Medicago sativa in italics.

Results

Line 121: Medicago sativa in italics.

Line 121: it is recommended to increase dimensions and improve quality - sharpness of Figure 1B.

Line 125: is color red?

Line 138: it is recommended to increase dimensions and improve the quality of Figure.

Line 141: it is recommended to increase dimensions and improve the quality of Figure.

Line 143-151: according to PCA how do these indicators correlate?

Line 147: should be... (Figure 4A)....

Line 150: it should be... PCA...

Line 164: it is recommended to increase dimensions and improve the quality of Figure 4A, B and C.

Line 172-173: it is recommended to initially make the description of Table 1 and then place the table. Check.

Line 179: should be ... (Figure 5A)...

Line 185: should be ... (Figure 5B)...

Line 187: check: ... et al. meanwhile... ??

Line 192-195: paragraph looks like a conclusion.

Line 197: it is recommended to increase dimensions and improve the quality of Figure 5A and B.

Line 213: ...jasmine?

Line 216: should be: ... Figure S1-2)....

Line 221: should be: ... Figure S3-7)...

Line 224: should be: ...(Figure 6D)...

Line 229: should be: ... Figure S6-9)...

Line 231: it is recommended to increase dimensions and improve the quality of Figure 6A, B, C and D....

Line 308: it is recommended to increase dimensions and improve the quality of Figure 7A and B.

Line 325: it is recommended to increase dimensions and improve the quality of Figure 8A and B.

Discussion

It is recommended to enrich this section by describing how this knowledge can improve crop productivity and quality.

Line 276: in italics ... Medicago sativa...

Line 293: should be... Figure 7)....

Line 314: should be... Figure 8...

Line 315: should be... Pisum sativum...

Line 320: should be: ...Arabidopsis thaliana's...

Line 328-330: In which plant is this enzyme mentioned?

Line 338: check ...CKIAA and SL...

Materials and methods

Line 344-346: ideally add averages of temperature, precipitation ...

Line 346-347: describe at what stage of alfalfa plant development the samples were collected. What parameters did you use to define the sampling time? How many samples were taken?

Line 350: define FAA.

Line 364: ¿?

Line 378: Yang et al[3]. Use...

Line 380: Check sentence...sequencing reads When the number...

Conclusions

Line 408: Medicago sativa in italics.

Line 408: I find Figure 9 presented by the authors very interesting, but I think it should be placed in the Results section.

References

Please check reference style.

Supplementary materials

Improve the quality of Figures.

Standardized language and font size of Tables.

Author Response

Dear reviewer:

We thank the reviewer for the kind consideration and constructive comments on our manuscript. We have carefully revised the manuscript and the changes in the revised manuscript have been highlighted. We hope these changes will strengthen our manuscript.

We would like to thank you for your professional review work, constructive comments, and valuable suggestions on our manuscript. As you are concerned, several issuesneed to be addressed, which are replied to in detail as below.

  1. We feel sorry for our carelessness. In our resubmitted manuscript, we have revised address instances of incorrect text formatting and punctuation usage throughout the manuscript. Thanks for your correction.
  2. We have revised the abstract to reduce its length in accordance with journal requirements (<200). Additionally, we have added more detailed descriptions of the experimental methods to enhance the clarity of our study's methodology.
  3. All images included in the manuscript have been submitted as high-resolution files for your review.
  4. according to PCA how do these indicators correlate?

In the PCA, the data matrix has been decomposed into principal components (PCs) that represent the orthogonal basis vectors, capturing the major variance directions within the dataset. The results indicate that PC1 accounts for 39.0% of the total variance, highlighting its significance in representing the primary trend across the dataset. PC2 further explains an additional 30.9% of the variance, suggesting that it captures another significant but distinct dimension of the data variability.

PC1: The indicators that contribute predominantly to PC1 are highly correlated with each other and represent a core set of variables that drive the major variance in the dataset. These indicators likely represent overarching phenomena or major experimental conditions influencing the dataset.

PC2: Indicators that load heavily on PC2, while still explaining a substantial portion of the variance, are likely orthogonal to those loading on PC1. These indicators are correlated among themselves but represent a different set of characteristics or conditions, potentially opposing or distinct from those delineated by PC1.

The remaining PCs (PC3-9), each account for progressively smaller portions of the total variance—9.2% for PC3, 7.7% for PC4, and so forth. These components elucidate finer gradations and more nuanced variations within the dataset. Indicators that correlate with these higher-order PCs tend to show weaker correlations with those associated with PC1 or PC2. While these components represent fewer dominant patterns, they may still provide potentially insightful details about the data, revealing subtle but important relationships or characteristics that are not captured by the first two principal components.

  1. Line 192-195: paragraph looks like a conclusion. 

We have revised this section to minimize discussion and focus more on describing the results.

  1. (Discussion) It is recommended to enrich this section by describing how this knowledge can improve crop productivity and quality. 

Thank you for your comprehensive and detailed suggestions. We have added content to the beginning of the Discussion section to enhance this part of our manuscript. Since our experiments are at a preliminary exploratory stage, we lack in-depth exploration and have not yet effectively integrated plant structure, material, and function in the context of practical applications. Your advice provides excellent guidance for designing our future experiments and improving our manuscript drafting process.

  1. In the Materials and Methods section, we have included additional details regarding the environmental conditions of the experimental site and the status of the plants at the time of sampling. These additions aim to further enhance the comprehensiveness of our experimental procedures.
  2. I find Figure 9 presented by the authors very interesting, but I think it should be placed in the Results section.

Thank you for your valuable feedback. We placed Figure 9 at the end of the article because our two experimental sections, although both focus on the stem base of alfalfa, are based on distinct approaches and do not directly relate. The first section explores the anatomical structure, while the second delves into metabolic substances and gene expression. Our intention in arranging the content this way is to provide a comprehensive summary that unites these distinct yet complementary aspects of our research. We aim to reinforce and deepen the reader's understanding of our study's theme by presenting a cohesive overview in the conclusion. We hope this clarifies our rationale for the structure of our manuscript. We sincerely appreciate your understanding and are grateful for your constructive feedback. Thank you for helping us improve our work.

    9.We have conducted a thorough review of the references to ensure the accuracy of the manuscript.

We sincerely appreciate your consideration of our manuscript. We look forward to your feedback and hope that our findings will contribute valuable insights to the field. Thank you for your time and effort in reviewing our work.

Sincerely,

Gao Qian

China Agricultural University

Reviewer 3 Report

Comments and Suggestions for Authors

The research to investigate the role of stembase of alfalfa as a connecting tissue between stems and roots and the sites of the bud generation is interesting. However, the manuscript needs some revisions as follows.

1) The regeneration of buds may depend on the seasons and harvesting. The authors should describe what status of the plants used for this experiment. It is desirable to study comparing the various types of stembase.

 2) In Figure 1, the photos of the the stem, stem base, and roots are shown. Please show the buds on the stembase, if present.

3) The area of the xylem was shown in red color, but the area of the phloem is not indicated. Please show the phloem part in the figure. 

4) The scientific name of alfalfa "Medicago sativa" should be expressed by italic. 

5) Materials and Methods: Please add more information about the plant at sampling.

Comments on the Quality of English Language

English is good.

Author Response

Dear reviewer:

We thank the reviewer for the kind consideration and constructive comments on our manuscript. We have carefully revised the manuscript and the changes in the revised manuscript have been highlighted. We hope these changes will strengthen our manuscript.

We would like to thank you for your professional review work, constructive comments, and valuable suggestions on our manuscript. As you are concerned, several issuesneed to be addressed, which are replied to in detail as below.

1) The regeneration of buds may depend on the seasons and harvesting. The authors should describe what status of the plants used for this experiment. It is desirable to study comparing the various types of stembase.

We have supplemented the Materials and Methods section to provide additional details regarding the plant materials used in this study, aiming to enhance the precision of our experimental results. The identification of "distinct stages of the stem base" represents significant research finding and serves as a focal point for our future research directions. We have already initiated some experiments in this regard and anticipate receiving further guidance from you in the future. We greatly appreciate your valuable feedback and guidance in this review process.

2) In Figure 1, the photos of the the stem, stem base, and roots are shown. Please show the buds on the stembase, if present.

We sincerely apologize for the oversight. During the sampling process of this experiment, we only captured overall images of the stem base, and detailed photographs of buds were not obtained. Unfortunately, it is not feasible to supplement this part with additional photos. We deeply regret this error on our part and any inconvenience it may have caused. Moving forward, we will make every effort to be more attentive to capturing details and focal points in our future experiments and research endeavors. Your understanding and forgiveness are greatly appreciated.

3) The area of the xylem was shown in red color, but the area of the phloem is not indicated. Please show the phloem part in the figure.

We have revised the images in this section by adding prominent annotations to highlight key points.

4) The scientific name of alfalfa "Medicago sativa" should be expressed by italic.

We genuinely appreciate the reviewer's meticulous examination of our work. Following their recommendations, we have rectified all relevant errors identified within the manuscript.

5) Materials and Methods: Please add more information about the plant at sampling.

Combining feedback from other reviewers, we have enhanced the "Materials and Methods" section in the manuscript to provide a clearer explanation of the experimental procedures.

We sincerely appreciate your consideration of our manuscript. We look forward to your feedback and hope that our findings will contribute valuable insights to the field. Thank you for your time and effort in reviewing our work.

Sincerely,

Gao Qian

China Agricultural University

Round 2

Reviewer 1 Report

Comments and Suggestions for Authors

Thanks for authors’ work, most of comments were well addressed, and the manuscript was well revised. I’m still worrying about some comments, (1) the qRT-PCR is missing, which would reduce reliability of the results. (2) figure 7 and 8 have still missing legends in pdf files, please check them.

Author Response

Dear reviewer:

We sincerely appreciate the time and effort invested by the reviewers in evaluating our manuscript.

After receiving the revised opinions of the first edition, we immediately prepared samples and carried out supplementary experiments. The content of qRT-PCR has been added to the article, and the newly added content includes the article and supplementary materials. It is hoped that this part can increase the reliability of our experimental results.

We have referred to some literature, and our pictures are basically consistent with them[1,2]. We think the pictures can express the main points of the article. I don't know which part of the comment needs to be added. Can you point it out so that we can make targeted modifications?

  1. Min, Z.; Zhao, X.; Li, R.; Yang, B.; Liu, M.; Fang, Y. Comparative Transcriptome Analysis Provides Insight into Differentially Expressed Genes Related to Bud Dormancy in Grapevine (Vitis Vinifera). Scientia Horticulturae 2017, 225, 213–220, doi:10.1016/j.scienta.2017.06.033.
  2. Min, Z.; Li, Z.; Chen, L.; Zhang, Y.; Liu, M.; Yan, X.; Fang, Y. Transcriptome Analysis Revealed Hormone Signaling Response of Grapevine Buds to Strigolactones. Scientia Horticulturae 2021, 283, 109936, doi:10.1016/j.scienta.2021.109936.

    Sincerely,

    Gao Qian

    China Agricultural University

Reviewer 2 Report

Comments and Suggestions for Authors

The authors addressed the suggestions/comments in a clear and concise manner. We believe that this manuscript has been improved and enriched.

Only verification:
* In the text, Medicago sativa L. is mentioned, but it should be Medicago sativa L.
* Verify font size in figure titles.
* Verify the format of the figure titles.
* Added space after the figure titles.

Author Response

Dear reviewer:

We sincerely appreciate the time and effort invested by the reviewers in evaluating our manuscript.

We sincerely thank the reviewer for careful reading. As suggested by the reviewer, we have corrected each image, also corrected the “Medicago sativa L.” into “Medicago sativa L”.

Sincerely,

Gao Qian

China Agricultural University

Reviewer 3 Report

Comments and Suggestions for Authors

The answers by the authors are appropriate, and the manuscript has been well-revised. The scientific name Medicago sativa L., only Medicago sativa should be written in italics. L is not italicized.

Author Response

Dear reviewer:

We sincerely appreciate the time and effort invested by the reviewers in evaluating our manuscript.

We have reconfirmed all the scientific names in the whole paper and made modifications. We were really sorry for our careless mistakes. Thank you for your reminder.

Sincerely,

Gao Qian

China Agricultural University
